# Thrombopoietin, the Primary Regulator of Platelet Production: From Mythos to Logos, a Thirty-Year Journey

**DOI:** 10.3390/biom14040489

**Published:** 2024-04-18

**Authors:** Kenneth Kaushansky

**Affiliations:** Renaissance School of Medicine, Stony Brook University, Stony Brook, NY 11794, USA; kenneth.kaushansky@stonybrook.edu

**Keywords:** hematopoiesis, platelet production, thrombopoietin, hematopoietic stem cell, myeloproliferative neoplasms

## Abstract

Thrombopoietin, the primary regulator of blood platelet production, was postulated to exist in 1958, but was only proven to exist when the cDNA for the hormone was cloned in 1994. Since its initial cloning and characterization, the hormone has revealed many surprises. For example, instead of acting as the postulated differentiation factor for platelet precursors, megakaryocytes, it is the most potent stimulator of megakaryocyte progenitor expansion known. Moreover, it also stimulates the survival, and in combination with stem cell factor leads to the expansion of hematopoietic stem cells. All of these growth-promoting activities have resulted in its clinical use in patients with thrombocytopenia and aplastic anemia, although the clinical development of the native molecule illustrated that “it’s not wise to mess with mother nature”, as a highly engineered version of the native hormone led to autoantibody formation and severe thrombocytopenia. Finally, another unexpected finding was the role of the thrombopoietin receptor in stem cell biology, including the development of myeloproliferative neoplasms, an important disorder of hematopoietic stem cells. Overall, the past 30 years of clinical and basic research has yielded many important insights, which are reviewed in this paper.

## 1. Introduction

In 1906, Carnot and Deflandre [1] established that red blood cell production was regulated hormonally by a substance they dubbed hemopoietine, a glycoprotein later termed erythropoietin. With this discovery, the field of blood cell production joined the endocrine systems of human physiology (e.g., glucose regulation, metabolic rate, blood pressure regulation, calcium metabolism) and set the stage for our current understanding of the most metabolically active organ of the body, marrow hematopoiesis. Unfortunately, the technology needed to advance our understanding of the method by which other cells of the blood are produced required an additional 60 years, when Leo Sachs in Israel and Don Metcalf (see Figure 1) in Australia independently established tissue culture techniques to grow colonies of granulocytes and macrophages [2,3]. Later, others established similar systems to grow lymphocytes, dendritic cells, mast cells and marrow megakaryocytes, the cells that shed platelets into the blood. It was quickly determined that these blood cell cultures require specific hematopoietic growth factors (HGFs) to develop from their marrow or blood progenitors: erythropoietin for red cells and what was termed a series of “activities” for each of the white cell types and platelets. The availability of colony-forming assays then allowed a number of the HGFs to be biochemically purified and molecularly cloned. Between 1977, with the purification of erythropoietin [4], and the early 1990s, blood cell growth factors for B and T lymphocytes (interleukin (IL)-7 [5]), neutrophils (granulocyte (G) colony-stimulating factor (CSF) [6]), monocytes (M-CSF [7]), eosinophils (IL-5 [8]), mast cells (IL-3 [9]), hematopoietic stem cells (HSCs), stem cell factor (SCF [10]) and dendritic cells (granulocyte-macrophage (GM)-CSF [11]) were defined, cloned and tested in animals and humans for therapeutic effects. Conspicuously absent from this golden age of HGF discovery was a growth factor for platelet production, dubbed thrombopoietin by Kelemen in 1958 [12]. Despite several claims of the purification of thrombopoietin, none of these efforts led to the cloning of the molecule, the sine qua non of HGF identification. Moreover, with the demonstration that administration of IL-6 or IL-11 to animals or humans could modestly (50% increase) stimulate platelet production, despite the fact that deletion of these cytokines failed to alter baseline platelet production, some felt that a distinct “thrombopoietin” was a myth, that it did not exist and rather that a combination of other cytokines likely control platelet production.

## 2. The Cloning of Thrombopoietin

Despite this skepticism, between June and November of 1994, four groups reported the purification, cloning and characterization of murine, rat, canine, porcine and human thrombopoietins.

Two strategies were employed by the four groups to clone the gene and/or cDNA for thrombopoietin. Two groups used multiple conventional chromatographic techniques and platelet-forming assays to purify the hormone from the plasma of multiple species of thrombocytopenic animals [13,14]. This Herculean conventional biochemical approach yielded microgram quantities of rat, dog and pig thrombopoietins. From the purified proteins, their N-terminal amino acid sequences were discerned and used to design nucleic acid probes to identify thrombopoietin clones from cDNA libraries.

A second approach was based on a prior discovery in a seemingly unrelated field of biology: the identification of a murine myeloproliferative leukemia virus (MPLV [15]). Following this seminal discovery, it was then realized that the viral, tumor-causing oncogene, *v-mpl*, represents a mutant form of a previously unknown gene present in virtually all mammalian species. First described in 1990 by Michele Souyri, this “protooncogene” was dubbed c-Mpl, and its human counterpart was cloned from the human erythroleukemia (HEL) cell line [16]. The primary amino acid sequence of c-Mpl displayed many hallmarks of several previously described cytokine receptors, such as the erythropoietin receptor, yet since its ligand was unknown, c-Mpl was termed an “orphan receptor”. Our laboratory hypothesized that *c-Mpl* encoded the thrombopoietin receptor, based on an epiphanic conversation I had in 1992 with my next-door laboratory neighbor, and friend, Thalia Papayannopolou (see Figure 2). A decade earlier, it was Thalia who developed the HEL cell line [17] from which a cDNA for *c-Mpl*, present at high levels, was initially cloned. As their name implies, HEL cells can be driven in tissue culture to resemble erythroid precursor cells. But what Thalia shared that day was that, under just the right conditions, HEL cells could also be coaxed to display a number of features of megakaryocytes, cells that give rise to all the blood platelets. Based on the logic that a cell that can display erythroid and megakaryocytic properties and bear high levels of a putative cytokine receptor that is not the erythropoietin receptor, we believed c-Mpl to be the receptor for the mythical thrombopoietin. We employed a functional expression cloning strategy to identify the receptor’s ligand. In our approach (KK in collaboration with scientists at Zymogenetics, Inc., Seattle, WA, USA), *c-Mpl* was introduced into BaF3 cells, a cell line absolutely dependent on exogenous IL-3 for survival and growth. The BaF3/c-Mpl-expressing cells were then subjected to chemical mutagenesis in order to (hopefully) induce autonomous, IL-3-independent growth instead, then dependent on mutagen-induced expression of a stimulatory ligand for c-Mpl. Several autonomously growing BaF3/c-Mpl clones developed, most autonomously expressing IL-3, but in two clones, a novel autocrine growth factor was expressed, subsequently cloned [18] and biologically characterized [19] as thrombopoietin. Others used the c-Mpl receptor as an affinity matrix to purify whatever its ligand turned out to be. In 1994, over a span of five months, we and three other groups reported the cloning and initial characterization of thrombopoietin [13,14,18,19,20].

## 3. The Interaction between Academia and Industry

As an aside, an “interesting thing” happened once the collaboration between myself and other University of Washington scientists with Zymogenetics yielded clones of thrombopoietin. Suddenly, all of our academic contributions to the effort (including the idea it would be possible to clone thrombopoietin, the hypothesis that c-Mpl was the thrombopoietin receptor, the use of BaF3/c-Mpl cells as a vehicle, the assays used to characterize the biological activities of the molecule, etc.) were suddenly felt by my colleagues at the company to be “obvious” and to be “simple” contributions that did not warrant being part of the claims of intellectual property. Such views were expressed to me, my scientific colleagues at my university and to my laboratory personnel. At first, this was quite perplexing, and unsettling, as our university/corporate collaboration had been so collegial and successful to that point. A conversation with Gene Goldwasser at a subsequent scientific meeting, who purified the erythropoietin with which AMGen was able to clone cDNA for the gene, put this “diminishing” into proper perspective. 

In 1980, the US Congress passed the Bayh/Dole Act (officially termed the Patent and Trademark Act Amendments), which allows universities and nonprofit research institutions to own, patent and commercialize inventions developed with federal funding. In essence, Bayh/Dole was designed to accelerate the translation of academic discovery into commercially successful (in our case, healthcare) products, by allowing a sharing of intellectual property between the university and the corporate entity. One result of this act was the amazing rise in offices of technology transfer in all the major universities in the land. A component of the Bayh/Dole Act was a provision that allowed the government a “piece of the action” if the academic contributor(s) was funded by the US Government (usually the NIH), which allowed the NIH to push the potential product into commercial use if the corporate entity chose not to do so. Hence, if the academic entity’s contribution was “trivial”, “obvious” in the language of patent attorneys, then the NIH would not be entitled to any say in the development of the intellectual property. It should be noted that in the 14 years between passage of the Bayh/Dole Act and our cloning of thrombopoietin, and in the additional 30 years since, the NIH has never prosecuted their rights outlined in the act. My conversation with Gene that day was quite surprising and enlightening, as no one at my university had shared these insights. I bring this forward as a “buyer beware” moment. There is little doubt that my collaboration with the company to clone thrombopoietin was incredibly fulfilling, but it also serves as a wake-up call to those academic investigators who choose to enter into collaborations with the commercial sector. Make sure all details are worked out in advance, so as to avoid “surprises”.

## 4. Biochemical Characterization of Thrombopoietin and Its Receptor

Biochemically, human thrombopoietin is a secreted protein composed of 332 amino acids which contains two functional domains. The amino terminal half of the full-length molecule has primary, secondary and tertiary structural homology to several other HGFs, in particular to erythropoietin. These and several other HGFs (e.g., GM-CSF, IL-5, IL-3, G-CSF) are characterized by a conserved disulfide structure and a four-α-helix-bundle fold in which the helixes display an up–down–up–down directionality.

This finding of a uniform protein fold for the HGFs had its origin in a remarkable report by Fernando Bazan [21], commenting on a paper reporting the crystal structure of IL-2 [22]. By reanalyzing the published electron density map present in the earlier paper, Bazan reasoned that several regions of low electron density were misassigned within the published tertiary structure of IL-2, and one could argue that instead of the authors initial conclusion that the orientation of the four α helices of IL-2 were oriented up–up–down–down, a novel protein fold, it could instead be surmised to be an up–down–up–down four-helix bundle, quite similar to the published tertiary structure of human growth hormone. The authors of the original IL-2 structure quickly agreed. Based on the newly identified concordance of the tertiary folds of IL-2 and GH, every subsequent tertiary structural solution of an HGF (EPO, GM-CSF, G-CSF, IL-3, IL-4, IL-5, IL-6, IL-7, etc.) aligns with this first family of structural homologues. More recent studies have confirmed this same tertiary fold characterizes thrombopoietin [23,24].

The biological activity of thrombopoietin is mediated entirely by the binding of the amino terminal 152 amino acids of the hormone (termed the receptor binding domain (RBD)) to the c-Mpl receptor. The thrombopoietin receptor is structurally homologous to a number of other cytokine receptors, especially the growth hormone and erythropoietin receptors [25]. The thrombopoietin receptor is a homodimer of two molecules of c-Mpl, as are the growth hormone, G-CSF, prolactin and erythropoietin receptors. In contrast, GM-CSF, IL-2 and other receptors are heterodimers or heterotrimers of distinct molecules. But like the thrombopoietin receptor, most of the known cytokine receptors are “monogamous”; i.e., the thrombopoietin receptor binds only to thrombopoietin, likewise for erythropoietin and its receptor. However, several cytokines that employ heterodimeric or heterotrimer receptors share one or two of the receptor subunits with other growth factors, respectfully, but nearly always have one devoted monogamous receptor subunit. For example, while the α chain of the heterodimeric GM-CSF receptor only binds to GM-CSF, its receptor β chain is also employed by the receptors for IL-3 and IL-5. Likewise for the IL-2 receptor; the α chain is IL-2 specific, but its β and γ chains are also employed by IL-7, IL-15 and other lymphocytic cytokine receptors [25].

Initially based on the analysis of the biological activities of series of mutated forms of several cytokines (e.g., see [25,26,27,28,29]), and subsequently based on co-crystal structures of several cytokine/receptor pairs, including that for erythropoietin, GM-CSF and thrombopoietin and their receptors [30,31], two faces of the growth factor, one composed of its A and D helixes, and one of its A and C helixes, engage the two molecules of the homodimeric erythropoietin or thrombopoietin receptors, or the heterodimeric receptors for GM-CSF, IL-3 or IL-5 [31]. Moreover, these receptors appear to multimerize upon ligand binding, necessary for signal transduction [31]. Another critical, shared structural feature of the HGF receptor family is that most employ a non-covalently bound, tethered intracellular kinase [32], initially termed “just another kinase” [33], more recently termed Janus (or JAK kinase) to initiate signal transduction.

## 5. Thrombopoietin Signaling

When thrombopoietin or one of its structurally related cytokines binds to its cognate receptor, the receptor undergoes a conformational change [34] resulting in the phosphorylation of tyrosine residues within its cytoplasmic domain, mediated by the tethered JAK kinase. These receptor phosphotyrosine residues then act as docking sites for a number of additional, secondary signaling molecules, which also become phosphorylated by JAKs. Most (but not all) of these sites then act to stimulate cell growth through secondary and tertiary messengers. While this paradigm is true for nearly all HGFs, two of the hematopoietic cytokine receptors (c-Kit (the SCF receptor) and Flt3 (the Flt3 ligand receptor)) contain a kinase subdomain in their intracellular domain, which performs the tyrosine phosphorylation events instead of a bound JAK kinase. While there are four known JAK kinases (JAK1, JAK2, JAK3 and TYK2), in the case of thrombopoietin, only JAK2 and TYK2 are employed by c-Mpl [35]. In contrast, JAK1 and JAK3 tend to be employed by the cytokines that control lymphopoiesis. The intracellular signaling molecules “downstream” of JAK2 and TYK2 employed by the thrombopoietin receptor include the mitogen-activated kinase (MAPK) pathway, the phosphoinositol-3-kinase (PI3K) pathway, the cAMP response element binding protein (CREB) pathway and two homeobox (Hox)-containing proteins, HoxB4 and HoxA9 ([36] and Figure 3).

## 6. The Relationship of Thrombopoietin and Other Hematopoietic Growth Factors

As noted above, like its HGF homologues, thrombopoietin contains a ~150 amino acid four-helix bundle domain which binds to its cognate receptor, c-Mpl. However, unlike all of its homologues, thrombopoietin also contains a second domain, composed of 183 carboxyl-terminal amino acids that bear several asparagine-linked sites of carbohydrate modification (which have a minor negative effect on its growth-promoting activities *in vitro*). The primary amino acid sequence of this carboxy-terminal domain does not resemble any known molecule, but functionally has been shown to greatly prolong the circulatory half-life of the RBD (P. Hunt, personal communication). Additional study of the C-terminal domain has shown it to be responsible for proper intracellular folding, and hence secretion of the amino-terminal RBD of thrombopoietin, a conclusion based on the following: (1) when expressed alone, the RBD is very poorly secreted from mammalian cells, and (2) the co-expression of the carboxyl-terminal domain as a separate molecule greatly enhances secretion of the RBD [38]. If you will, the carboxyl-terminal domain acts as an intramolecular chaperone for the proper folding of the thrombopoietin RBD.

While displaying some unique molecular characteristics compared to its structural homologues (e.g., the carboxyl-terminal, “intramolecular chaperone” domain), as discussed above the secondary and tertiary structure, the structure–function relationships of regions of the molecule and the mechanisms of cellular signaling of thrombopoietin are quite similar to most/all of the other HGFs. In contrast, the *range* of the biological activities of thrombopoietin is unlike most of the other HGFs.

## 7. The Biological Activities of Thrombopoietin

While initially predicted to act primarily or solely as a differentiation factor for the megakaryocytic lineage, i.e., the primary cytokine supporting terminal platelet formation, the hormone was not initially thought to promote expansion of platelet progenitor cells, marrow megakaryocytes (MKs). Our first surprise when characterizing the biological activity of thrombopoietin was the discovery of its potency as a growth factor for MK progenitor cells, the so-called colony-forming unit (CFU)—MK [19]. By itself, thrombopoietin is the most potent megakaryocytic colony-stimulating factor known. In fact, the presence of thrombopoietin in terminal cultures of platelet-producing megakaryocytes was reported to inhibit platelet formation [39] as the withdrawal of thrombopoietin stimulates platelet formation in tissue culture. Our second biological surprise was that in addition to expanding MK progenitors, thrombopoietin also acts in concert with other HGFs to expand erythroid and myeloid progenitors [40]. And this effect was likely mediated by its third surprising feature; thrombopoietin acts to promote the survival and, in the presence of SCF, expansion of HSCs in culture [41,42]. This unexpected conclusion was later supported by (1) the finding of c-Mpl expression on primitive hematopoietic cells [43]; (2) the fact that genetic elimination of c-Mpl or thrombopoietin in mice results in a 7–8-fold reduction in transplantable HSCs [44]; and (3) “experiments of nature”; genetic mutation or elimination of thrombopoietin, or more commonly of c-Mpl, in humans leads not only to greatly reduced numbers of megakaryocytes and platelets (an inherited disorder termed amegakaryocytic thrombocytopenia), but also to aplastic anemia in affected children [45,46]. Hence, despite its name, thrombopoietin is also a stem cell factor.

As noted previously [36], a myriad of signaling mediators are employed by the thrombopoietin/thrombopoietin receptor complex to drive its biological effects on cells ranging from the hematopoietic stem cell to megakaryocytic progenitor cells. Very recent studies indicate that these are not “all or nothing” responses; that is, the effects on stem cells appear to depend on pathways different than the effects on megakaryocytic progenitor cells. As also noted previously, the tertiary structure of thrombopoietin bound to its receptor was recently determined by cryoelectron microscopy [30]. These investigators then went on to use the structure to design thrombopoietin antagonists, superagonists and partial agonists that can differentially affect stem cells vs. megakaryocytic progenitor cells. Remarkably, these cell-type-specific partial agonists employ different intracellular pathways for their biological effects. These conclusions have multiple implications for the basic biology of how growth factors function, and on how to create designer molecules to generate “more desirable” effects. As will be discussed below, the role of the thrombopoietin receptor in myeloproliferative neoplasms has driven the search for receptor antagonists and the need to expand hematopoietic stem cells in vitro for a myriad of ex vivo cellular applications, only two of which include expansion for transplantation and manipulation to correct genetic mutations (e.g., sickle hemoglobin) and are but two practical applications of the approach illustrated in the work of Tsutsumi and colleagues [30].

## 8. The Thrombopoietin Receptor Is Involved in Myeloproliferative Neoplasms

As noted above, our current understanding of thrombopoietin biology was catalyzed by the realization that the thrombopoietin receptor is encoded by the protooncogene *c-Mpl*, which when mutated (to *v-mpl*) leads to myeloproliferative leukemia in mice [15]. A number of more recent observations of c-Mpl expression and function illustrate the importance of the wild-type receptor in the biology of *human* myeloproliferative neoplasms (MPNs). The first inkling that wild-type c-Mpl might be linked to MPNs, disorders of HSCs, came from a paper published by Li and colleagues in 1996 [47]. In this report, the authors found that a reduction in c-Mpl expression in marrow cells from patients with MPNs reduced “endogenous colony formation”. One of the hallmark features of human MPNs is that in contrast to normal marrow cells, which require physiological levels of exogenous HGFs for survival and proliferation, marrow and blood stem and progenitor cells from patients with MPNs, which include polycythemia vera (PV), primary myelofibrosis (PMF) or essential thrombocythemia (ET), form blood cell colonies in the apparent absence of HGFs. This seminal observation regarding “endogenous erythroid colony formation”, i.e., erythroid colony formation in the absence of erythropoietin, was made by Jeff Prchal in 1974 [48]. In reality, in contrast to normal hematopoietic progenitor cells, the growth of hematopoietic stem and progenitor cells from patients with MPNs is not truly independent of the primary growth factor for that lineage, but rather is exquisitely sensitive to the growth-promoting activity of erythropoietin [49], or in the case of myeloid and megakaryocytic colony-forming cells, the corresponding growth factors [50], whose levels are linked to the small amounts of serum in such cultures. In 1996, after the c-Mpl protooncogene was described, Li and colleagues demonstrated that endogenous colony formation in marrow cells from patients with MPNs was significantly inhibited by down-modulation of c-Mpl expression. Shortly thereafter, we proposed, based on this observation, that c-Mpl might be involved in human MPNs [51,52]. Additional circumstantial evidence for this hypothesis has come from a number of sources, including (1) expression of c-Mpl on human HSCs [43], the cell of origin of MPNs; (2) aberrant expression of wild-type c-Mpl in mice induces an MPN [53]; and (3) c-Mpl is overexpressed on the neoplastic cells of about half of patients with acute myelogenous leukemia and myelodysplastic syndromes [54]. And finally, since c-Mpl is the only JAK2-dependent HGF receptor expressed in HSCs; since expression of the oncogene JAK2V617F is responsible for disease in virtually all patients with PV and half with PMF and ET [55]; and since all JAKs must bind to an HGF receptor to function, c-Mpl is again implicated in human MPNs. Several other groups have also come to the conclusion that c-Mpl is involved in human MPNs, including those initiated by other oncogenes [56]. But a definitive conclusion that c-Mpl is required for the development of MPNs awaited direct experimental proof in a model MPN.

## 9. A Mechanism by Which c-Mpl Participates in Hematopoietic Neoplasms

As discussed above, initiation of thrombopoietin signaling by c-Mpl is dependent on the cytoplasmic kinase JAK2. In 2005, four groups (somewhat) independently demonstrated that mutation of JAK2 to JAK2V617F is found in the vast majority of patients with PV and half of patients with the other two major MPNs, ET and PMF [55,57,58,59]. A number of investigators also showed that introduction of a JAK2V617F transgene into HSCs induces an MPN in mice [55,60]. Using a transgenic model of an MPN [60], we then showed that elimination of c-Mpl in HSCs prevents the development of a JAK2V617F-induced murine MPN [61]. Moreover, more recently, a second group of mutations associated with MPNs, C-terminal truncations of calreticulin, has been shown to transform cells by binding to c-Mpl [56,62,63]. Thus, down-modulation of c-Mpl has been suggested as a potentially promising treatment approach for patients with MPNs [56,64]. Finally, although quite rare, mutation of c-Mpl itself can lead to an MPN [65]. This subject was recently reviewed [66] and is illustrated on the right side of Figure 3.

## 10. The Clinical Uses of Thrombopoietin and Thrombopoietin Receptor Agonists

Finally, the identification, cloning and characterization of thrombopoietin as the primary regulator of platelet production, as well as a critical effector of HSC biology, have led to its clinical development for patients with thrombocytopenia of several origins, as well as aplastic anemia. A brief history of the initial efforts to utilize thrombopoietin in the clinic will be presented here, as it is very instructive on several levels.

In 1996, two years after the cloning of thrombopoietin, clinical trials of the hormone were initiated by both AMGen and Kirin Pharmaceuticals. The version of thrombopoietin employed by AMGen was a highly engineered molecule produced in bacteria, containing only the RBD and modified with several sites of polyethylene glycol to improve its circulatory half-life, termed megakaryocyte growth and development factor (MGDF). Therapeutic clinical trials of MGDF were performed in healthy, potential platelet donors who were given several subcutaneous doses of the drug at three-week intervals in order to bolster their platelet counts prior to apheresis for platelet transfusion into thrombocytopenic individuals. Additional clinical trials were launched in patients undergoing myelosuppressive chemotherapy for malignancy. In contrast, Kirin produced full-length thrombopoietin in mammalian cells for intravenous administration in clinical trials in thrombocytopenic patients. While the patients in the AMGen clinical trials responded to MGDF with substantial thrombocytosis (e.g., 3-fold increased platelet counts in normal volunteers), and subsequent apheresis yielded impressive platelet collections, following the multiple dosing regimen, a number of the platelet donors began developing severe thrombocytopenia due to the development of anti-MGDF antibodies that cross reacted with the recipients’ endogenous thrombopoietin [67,68]. Within the same year, the clinical trial was terminated and MGDF was never again administered to humans in the US. In contrast to this life-threatening complication with MGDF, an intravenous Kirin full-length, native thrombopoietin was well tolerated, stimulating thrombopoiesis in recipients, but without developing any evidence of anti-drug or anti-native thrombopoietin antibodies. Nevertheless, all clinical use of any form of full-length or other engineered version of the native hormone was abandoned in the US, although the drug has been used successfully to treat immune thrombocytopenia (ITP) and chemotherapy-induced thrombocytopenia in China. The lesson from this early clinical application is “it’s not nice to fool with mother nature”.

In response to this pharmaceutical failure, several routes to develop small molecular mimics of thrombopoietin were catalyzed and have been very successful in the US and Europe; these molecules are now termed thrombopoietin receptor agonists (TRAs). At present, there are four TRAs approved for use in various thrombocytopenic clinical settings by both the US Food and Drug Administration (FDA) and the European Medicines Agency (EMA): romiplostim, eltrombopag, avatrombopag and lusutrombopag.

Romiplostim was the first TRA approved for clinical use in 2008. Romiplostim is a “peptibody”, containing four copies of a small peptide selected to bind to and stimulate the thrombopoietin receptor, all attached to an immunoglobulin constant region scaffold [69], which provides a long circulatory half-life to the engineered molecule. Of note, the RBD-binding peptides bear no amino acid homology with thrombopoietin, obviating the production of antibodies that bind to the endogenous hormone. As the drug is essentially an immunoglobulin, it must be administered parenterally. The initial clinical trials of romiplostim were conducted in patients with refractory immune thrombocytopenia (ITP), defined as patients who have failed or have had inadequate responses or severe complications to numerous agents (e.g., glucocorticoids, splenectomy, intravenous immunoglobulin, rituximab). Depending on whether the patients were post-splenectomy or were spleen replete, the favorable response rate to romiplostim was 80–90% [70], defined as reduced or eliminated bleeding complications and an increase in platelet counts to above 50 × 10^9^/L, which was usually maintained as long as the patient remained on the drug. Moreover, follow-up studies have indicated that up to 30% of patients treated for 2 years can be successfully weaned from the drug and retain an adequate platelet count [71], a somewhat unexpected finding as the drug was not thought to be disease-altering like the various immune-modulating therapies which served as the mainstay of treatment for ITP prior to the development of TRAs. A recent expert panel has provided some advice on how/when to taper TRAs in patients with ITP [72].

The second TRA licensed for use in thrombocytopenia, eltrombopag, is a small organic molecule (M_r_ 442) that is orally active, identified by screening huge libraries of organic compounds for those that could stimulate the growth of c-Mpl-bearing cells [73]. Clinical trials of eltrombopag were conducted in patients with refractory ITP soon after those of romiplostim, and were similarly successful in long-term follow-up studies correcting severe thrombocytopenia [74], with only minimal side effects (a small fraction of patients developed a mild increase in liver function tests). Eltrombopag was approved by the FDA and EMA soon after romiplostim. Moreover, because of the effects of thrombopoietin on HSCs, clinical trials of the drug were also conducted in patients with severe aplastic anemia (SAA). Compared to the standard of care for patients without a marrow transplant donor (i.e., severe immunosuppression), the addition of eltrombopag to immunosuppressive therapy was associated with markedly higher rates of hematologic response [75], leading to FDA approval of the drug for SAA.

Avatrombopag and lusutrombopag are second-generation small organic molecules that bind to and activate the thrombopoietin receptor, which were identified in a similar fashion to that of eltrombopag. Both agents have been shown to increase platelet production in patients who are thrombocytopenic due to chronic liver disease. Both drugs have now been approved by the FDA and the EMA for patients with liver disease and thrombocytopenia requiring a medical procedure that carries a significant risk of bleeding [76,77]. Avatrombopag was also effective for patients with ITP who needed to discontinue romiplostim or eltrombopag [78]. In a recent, small, head-to-head comparison to eltrombopag in patients with SAA, patients treated with avatrombopag displayed faster platelet recovery and had a higher platelet level in the second month of therapy than those treated with eltrombopag, as well as a lower rate of side events [79].

Finally, while there is great need for a potent agent to moderate or alleviate chemotherapy-induced thrombocytopenia (CIT), and there are numerous reports of some benefits [80], unfortunately there are no definitive, rigorous clinical trials establishing the efficacy of TRAs in such settings [81] at the time of writing of this paper.

The cost of all four TRAs is high. For example, a weekly dose of romiplostim costs approximately USD 1000, not counting the clinical testing and administrative costs. For most patients with chronic refractory ITP, treatment must continue indefinitely in order for the beneficial effects of the drugs to be maintained. Thus, based on the results of a six-strategy Markov model to evaluate the most cost-effective approach to refractory ITP, many experts recommend that such patients first be tried on immune-modulating therapy with splenectomy and/or rituximab [82]. However, should such treatment fail, or the patient be intolerant to that approach, recent cost–benefit analyses have suggested that when used to treat patients with refractory ITP, TRAs display a reasonable cost profile [80]; other studies suggest that eltrombopag may be preferred over romiplostim because of the lower drug costs of the former [83].

Initially, it was felt that the use of thrombopoietin and TRAs in patients with ITP would not be disease-modifying, but rather simply a mechanism to drive increased platelet production and, in so doing, overcome the platelet (and megakaryocyte) destruction caused by the platelet/megakaryocyte antigen-directed autoimmune phenomenon. Thus, a patient with chronic ITP would theoretically need TRA therapy for their lifetime. With the widespread use of TRAs in patients with refractory, chronic ITP came the discovery that some patients could be successfully weaned from the TRA while maintaining adequate drug-free platelet counts [84]. Overall, it is estimated that about one-third of patients with refractory ITP can be so weaned, although a recent consensus conference could not determine the predictors of successful discontinuation, monitoring intervals or rates of successful discontinuation or relapse [85].

Over the past 30 years, thrombopoietin has moved from myth to logic, from an “activity” (platelet formation) that many believed did not reside in a unique molecule, to its cloning, biological and structural characterization, to its role in normal and neoplastic hematopoiesis, and to its widespread use to treat several life-threatening conditions including severe aplastic anemia or thrombocytopenia due to autoimmunity, chemotherapy-related toxicity and liver failure. The search for a safe clinical agent capable of substituting the full-length hormone led to the discovery of several thrombopoietin receptor agonists, an accomplishment that serves as a poster child for the rational development of small, orally active drugs to replace expensive or difficult-to-produce therapeutic proteins. Moreover, the cytokine serves to point out that “you cannot always judge a book by its cover”. Thrombopoietin was supposed to be the effector of platelet production, nothing more. In fact, more careful study revealed it to be one of only two indispensable keepers of the hematopoietic stem cell. And finally, another logical lesson learned from the discovery of thrombopoietin 30 years ago is that sometimes, a finding in one area of science (murine viral oncogenesis) can be catalytic for a seemingly unrelated scientific or medical conundrum, which is, in this case, the myth that was thrombopoietin.

## Figures and Tables

**Figure 1 biomolecules-14-00489-f001:**
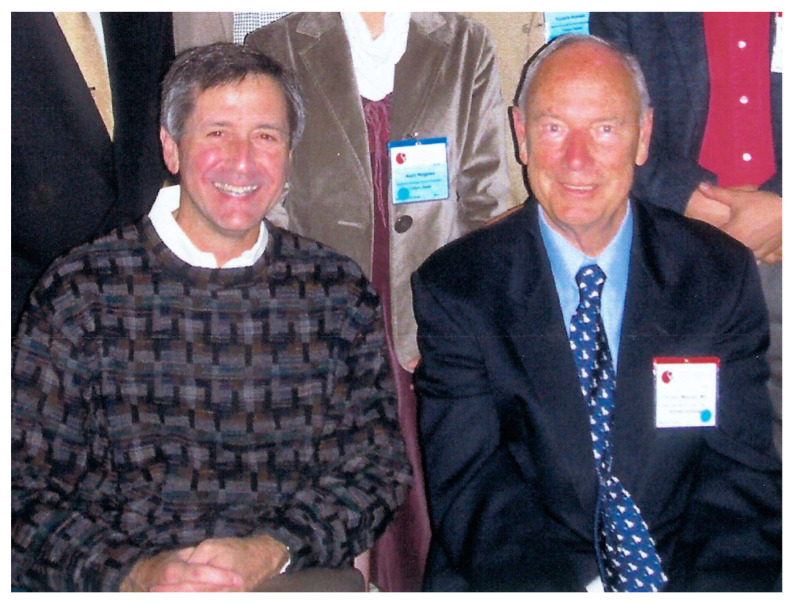
Don Metcalf (**right**) along with the author (**left**) attending the 46th annual meeting of the American Society of Hematology, December 2004, in San Diego, CA, USA.

**Figure 2 biomolecules-14-00489-f002:**
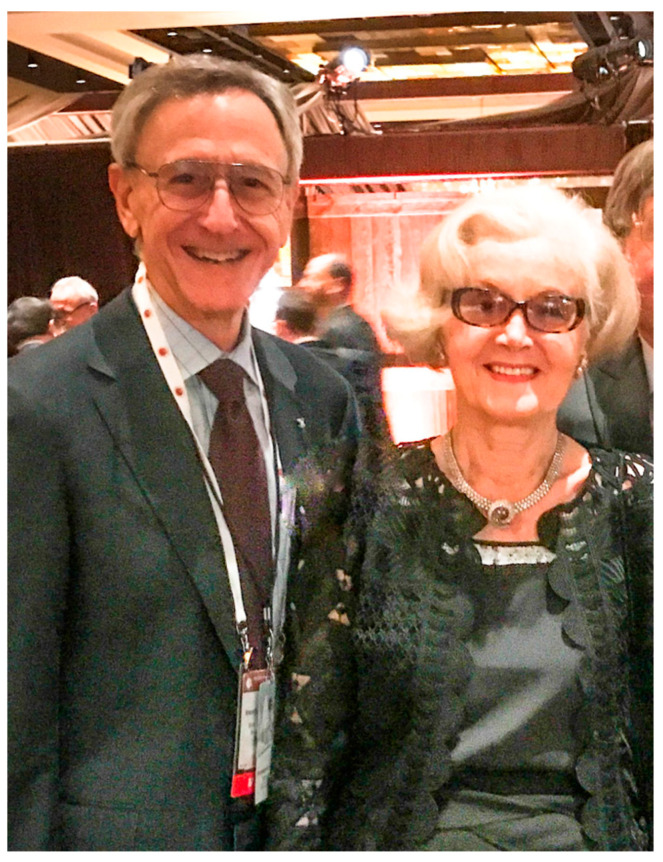
Thalia Papayannopoulou (**right**) along with the author (**left**) attending the 46th annual meeting of the American Society of Hematology, December 2019, in Orlando, FL, USA.

**Figure 3 biomolecules-14-00489-f003:**
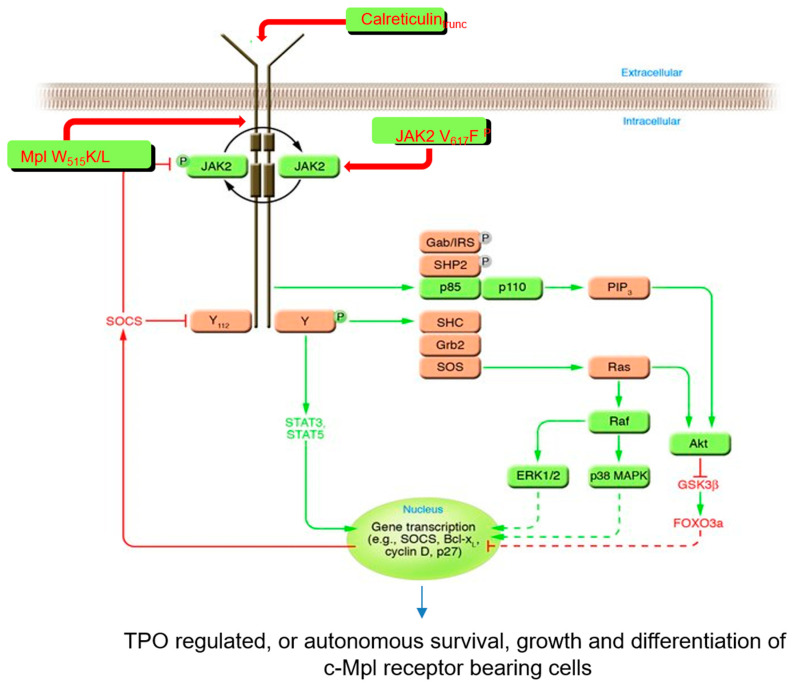
A partial “circuitry diagram” of the signaling pathways stimulated by the binding of thrombopoietin to its cell surface receptor. The molecules that signal the presence of thrombopoietin bound to c-Mpl are shown in colored blocks. Those molecules that are constitutively active in the presence of activating mutations associated with myeloproliferative neoplasms (JAK2V_617_F, C-terminal truncation mutants of calreticulin and c-MplW_515_K or W_515_L). Reprinted and revised with permission from Kaushansky K. [37].

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
