# Peer review of "Thrombopoietin, the Primary Regulator of Platelet Production: From Mythos to Logos, a Thirty-Year Journey"

_biomolecules, 2024, doi:10.3390/biom14040489_

Round 1

Reviewer 1 Report

Comments and Suggestions for Authors

The veracity of statements related to ref#18 should be checked.

Author Response

All of the statements regarding current ref 18 are well accepted, this was our original TPO cloning paper which has been cited countless times…it is possible the reviewer looked at the initial manuscript, before my adding refs (as requested by another reviewer) and the ref 18 is the paper on the cloning of c-Mpl from HEL cells, which did need to be corrected as indicted by another reviewer

Reviewer 2 Report

Comments and Suggestions for Authors

In this study “Thrombopoietin, the Primary Regulator of Platelet Production: From Mythos to Logos, a Thirty-Year Journey ” by Kenneth Kaushansky, the author has provided a detailed account of the early research into the biology of thrombopoietin, its subsequent cloning, purification and characterization as one of the critical hematopoietic growth factor, and finally has discussed its importance in the clinical conditions of defective hematopoiesis and various neoplastic conditions. Overall, this is a very informative piece of review, which at a glance gives all the necessary information about thrombopoietin to the readers who are not even familiar with it. The author is an accomplished researcher in the field of thrombopoiesis research and have provided some historical and personal experiences about not only research, but also about the intricacies of working with pharmaceutical companies to commercialize the research products, which is certainly informative and helpful for other researchers. The review of thrombopoietin, its receptor c-Mpl, their similarities and dissimilarities with other hematopoietic growth factors, signalling downstream to the receptor ligation, their role in normal physiology and in various hematopoietic clinical conditions and neoplasms, and the development of various thrombopoietin agonists and antagonists for clinical use are the strengths of this article. However, there remain some concerns that make the review less interesting to read by general readers. There seems to be a particular lack of attention to the details. The author should look into the concerns to make the review more appropriate for the readers.

1.      The major concern is the lack of sub-headings in the text. It feels like reading a long text. As the author provides the historical background to thrombopoietin research followed by subsequent discoveries and finally the update on current situation in the field, it could have been easily described in distinct sections with appropriate sub-headings.

2.      A diagram or scheme summarizing the review (showing the signalling molecules, biological effects and association with diseases etc.) would have been very helpful for the readers.

3.      Thrombopoietin’s  involvement in hematopoietic stem cell biology could have been elaborated a bit more in a separate section.

4.      In lines, 11, 14, 69 and 166, there are errors, which need to be rectified.

5.      References cited are not uniform in style. In lines 178 and 181 the references should be numbered as it is throughout the text.

6.      The sentence in lines 196-197 is incomplete.

7.      In lines 246-47, ‘in humans’ is used twice, which is not needed.

8.      In lines 295-303, the sentence is very long and complex to read. Better to make it simple with small sentences.  

Comments on the Quality of English Language

Moderate editing is required.

Author Response

  1. The major concern is the lack of sub-headings in the text. It feels like reading a long text. As the author provides the historical background to thrombopoietin research followed by subsequent discoveries and finally the update on current situation in the field, it could have been easily described in distinct sections with appropriate sub-headings. This has now been done                                                                                                  

  1. A diagram or scheme summarizing the review (showing the signalling molecules, biological effects and association with diseases etc.) would have been very helpful for the readers. This has been added as Figure 3

  1. Thrombopoietin’s involvement in hematopoietic stem cell biology could have been elaborated a bit more in a separate section. The review is already quite long, and so I have added some clinical citation of its stem cell effect in aplastic anemia

  1. In lines, 11, 14, 69 and 166, there are errors, which need to be rectified. Addressing this comment is now impossible as the line numbers are all changed…I have gone over the paper many times, making many corrections so I believe it likely these “errors” are now addressed

  1. References cited are not uniform in style. In lines 178 and 181 the references should be numbered as it is throughout the text. Completed, and I thank the rveiwr4 for picking this up
  2. The sentence in lines 196-197 is incomplete. Please see comments to question 4
  3. In lines 246-47, ‘in humans’ is used twice, which is not needed. Agree, corrected
  4. In lines 295-303, the sentence is very long and complex to read. Better to make it simple with small sentences. Agree, have divided the sentenced up and it reads easier

Reviewer 3 Report

Comments and Suggestions for Authors

In this current review “Thrombopoietin, the Primary Regulator of Platelet Production: 2 From Mythos to Logos, a Thirty-Year Journey”. The author reviews about the journey of Thrombopoietin which is well articulated and is told as story in order to capture the audience interest, few things can be added and the article can be published.

1)In line 11 and 14 instead of e in hematopoietic and e in engineered is written as 4, author need to correct that.

2) Resolution of Fig. 3 should be improved.

Author Response

In this current review “Thrombopoietin, the Primary Regulator of Platelet Production: 2 From Mythos to Logos, a Thirty-Year Journey”. The author reviews about the journey of Thrombopoietin which is well articulated and is told as story in order to capture the audience interest, few things can be added and the article can be published.

1) In line 11 and 14 instead of e in hematopoietic and e in engineered is written as 4, author need to correct that. Thanks, I missed that and its now corrected

2) Resolution of Fig. 3 should be improved I have tried but could not, as it was taken from an older slide of mine  with this note I ask the publishers to attempt to do this

Round 2

Reviewer 2 Report

Comments and Suggestions for Authors

The revised manuscript has improved significantly after the introduction of the sub-headings. However, the author seems to be not very receptive to the other comments and have acted half-heartedly or not at all.

1.      The spelling error in line 69 of original version (73 in the revised version) has not been corrected.

2.      In line 166 (revised 173) ‘binds’ has been used twice and despite pointing it out it has not been corrected.

3.      The sentence in lines 196-197 (revised 205-207) is still the same and incomplete. No correction here.

The author’s response ‘Addressing these comments is now impossible as the line numbers are all changed…I have gone over the paper many times, making many corrections so I believe it likely these “errors” are now addressed’ does not sound convincing and indicates carelessness. It will take hardly any time to locate these lapses and rectify them.

4.      Most importantly, the author has not worked on the comment #2. A scheme/model showing the essence of the entire review is not there. The author has just reprinted a nearly 25 years old figure from his previous publication. Although it shows the signalling molecules, it fails to show the various biological effects of thrombopoietin and the diseases associated with dysregulated thrombopoietin signalling. The 2005 figure needs an update for this review.

5. In revised line 312, it should be 'And finally......'

Comments on the Quality of English Language

Minor modifications necessary as outlined in my comments.

Author Response

I have now corrected nearly all of the spelling errors/repeated word/incomplete sentence, requests that were mentioned in the new review from reviewer 2, and have revised the figure which may or may not satisfy reviewer 2 

I have revised figure 3 to include a second section which illustrates the biology of c-Mpl in myeloproliferative neoplasms, which I think is at least some of what the reviewer wants.  You should know that as I have retired from my university, I have no ability to do medical illustrations, and hence, I have revised figure 3 the best I can.

All of the misspelled words, the repeated word, the long sentence has been divided into two sentences and the incomplete sentence has been completed….I thank the reviewers who found these errors.

Moreover, in making this final revision, I have included additional recent references that speak to the unique biology of this molecule, which was a critique from several reviews ago, which I did address with some new references, but not there are a few more (e.g. 63 and 64)